# Concentration-Dependent Pro- and Antitumor Activities of Quercetin in Human Melanoma Spheroids: Comparative Analysis of 2D and 3D Cell Culture Models

**DOI:** 10.3390/molecules26030717

**Published:** 2021-01-30

**Authors:** Harald Hundsberger, Anna Stierschneider, Victoria Sarne, Doris Ripper, Jasmin Schimon, Hans Peter Weitzenböck, Dominik Schild, Nico Jacobi, Andreas Eger, Josef Atzler, Christian T. Klein, Christoph Wiesner

**Affiliations:** 1Department of Medical and Pharmaceutical Biotechnology, IMC University of Applied Sciences Krems, A-3500 Krems, Austria; harald.hundsberger@fh-krems.ac.at (H.H.); anna.stierschneider@fh-krems.ac.at (A.S.); Victoria.sarne@meduniwien.ac.at (V.S.); doris.ripper@fh-krems.ac.at (D.R.); jasmin_schimon@hotmail.com (J.S.); hans.weitzenboeck@fh-krems.ac.at (H.P.W.); dominik.schild@fh-krems.ac.at (D.S.); andreas.eger@fh-krems.ac.at (A.E.); christian.klein@fh-krems.ac.at (C.T.K.); 2Department of Dermatology, Paracelsus Medical University, A-5020 Salzburg, Austria; 3Department Life Sciences, Research Institute for Applied Bioanalytics and Drug Development, IMC University of Applied Sciences Krems, A-3500 Krems, Austria; jacobi.nico@gmail.com; 4Molecular Devices, LLC, A-5071 Wals-Siezenheim, Austria; josef.atzler@moldev.com

**Keywords:** quercetin, phenotypic drug screening, melanoma, 3D spheroids, Nrf2 signaling

## Abstract

Quercetin, a dietary flavonoid found in fruits and vegetables, has been described as a substance with many anti-cancer properties in a variety of preclinical investigations. In the present study, we demonstrate that 2D and 3D melanoma models exhibit not only different sensitivities to quercetin, but also opposite, cancer-promoting effects when metastatic melanoma spheroids are treated with quercetin. Higher concentrations of quercetin reduce melanoma growth in three tested cell lines, whereas low concentrations induce the opposite effect in metastatic melanoma spheroids but not in the non-metastatic cell line. High (>12.5 µM) or low (<6.3 µM) quercetin concentrations decrease or enhance cell viability, spheroid size, and cell proliferation, respectively. Additionally, melanoma cells cultivated in 2D already show significant caspase 3 activity at very low concentrations (>0.4 µM), whereas in 3D spheroids apoptotic cells, caspase 3 activity can only be detected in concentrations ≥12.5 µM. Further, we show that the tumor promoting or repressing effect in the 3D metastatic melanoma spheroids are likely to be elicited by a precisely controlled regulation of Nrf2/ARE-mediated cytoprotective genes, as well as ERK and NF-κB phosphorylation. According to the results obtained here, further studies are needed to better characterize the mechanisms of action underlying the pro- and anti-carcinogenic effects of quercetin on human melanomas.

## 1. Introduction

Malignant melanomas are severe forms of cancer and are responsible for more than 75% of skin cancer deaths. In the last 50 years, their incidence has risen faster than almost any other cancer worldwide [1]. This increase in incidence is alarming, but what is also alarming is the very high propensity of melanomas to metastasize and the inefficiency of the current systemic treatments [2]. Once a melanoma becomes metastatic, the 5-year survival with treatment is approximately 15%–20%, and the median survival ranges from 8 to 12 months [3,4].

Many naturally occurring dietary compounds found in fruits and vegetables, such as quercetin, curcumin, resveratrol, epigallocatechin-3-gallate, lycopene, etc., have been well recognized as sources for drugs with cancer-preventive and/or anti-cancer properties [5]. Quercetin (3,3′,4′,5,7-pentahydroxyflavone) is the major flavonoid in the human diet, with an estimated daily dietary intake between 5 mg and 40 mg, although these levels can increase to 200–500 mg/day in individuals who consume fruits and vegetables particularly rich in this compound, such as tea and/or red wine [6]. Quercetin is described to have no or a very low toxicity in humans when administered orally in single doses of 4 g or 500 mg thrice daily [7,8]. The beneficial effects of quercetin in normal tissues have been mainly attributed to the excessive scavenging activity against reactive oxidative species (ROS), which consequently reduces DNA damage and ROS-associated carcinogenesis. Besides these anti-oxidative qualities, quercetin has also been reported to show pro-oxidative activity, which is dependent on its plasma and tissue concentrations, bioavailability, metabolism, and the time of exposure [9]. Low concentrations (<40 µM) have been reported to show antioxidant activities, whereas high concentrations exert the opposite effect via the oxidation of quercetin into o-quinone and the formation of ROS [10]. These pro-oxidative properties have been described as the main reason for the anti-tumor activity of quercetin in different cancers, including melanoma [11,12]. In contrast, low doses of quercetin enhance the total antioxidant capacity of cancer cells and antagonize the toxic effect of anti-neoplastic drugs [9,13].

Nuclear factor erythroid 2-related factor 2 (Nrf2) is considered to be the major cellular defense mechanism against oxidative stress through the controlled activation of a series of genes, including phase-II detoxifying enzymes, endogenous antioxidants, and transporters that protect cells from the destructive effects of carcinogens and environmental toxins [14]. Under normoxic conditions, Nrf2 is bound by Kelch-like ECH-associated protein 1 (Keap1) and persists in an inactivated status through ubiquitination and degradation in the proteasome [15]. Oxidative stress causes conformational changes as a result of the oxidation of thiol-sensitive amino acids in the Keap1-Nrf2 complex and the dissociation of Nrf2 from Keap1. Nrf2 then translocates into the nucleus, where it transcribes antioxidant response element (ARE)-dependent genes in order to balance oxidative mediators and maintain cellular redox homeostasis [16]. While transient Nrf2 activation in response to stress is described to have beneficial effects, persistently high levels of Nrf2 may, in contrast, protect cancer cells against the cytotoxic effect of chemotherapeutic agents and foster aggressive tumorigenesis in cancer cells [17]. Quercetin has been attributed to many positive effects, which have been described mainly in in vitro studies in various human malignant cell lines. It is recommended to be developed as an anti-cancer compound for use as either a preventative or therapeutic agent [18,19].

In order to study different dietary polyphenols and their therapeutic effects in human metastatic melanoma cell lines, we have developed a simple, fast, and reliable high-content screening (HCS) platform to analyze the anti-carcinogenic effects in both 2D and 3D cultures. These melanoma models not only show differences in terms of substance sensitivity but also the reverse, cancer-promoting effects, when metastatic melanoma spheroids are treated with quercetin. The present work shows that the pro- and anti-carcinogenic effects of quercetin are concentration-dependent and that there are sensitivity differences between the 2D and 3D models.

## 2. Results

### 2.1. Growth and Morphological Characteristics of 2D Cell Cultures versus 3D Melanoma Spheroids

Two-dimensional (2D) cell models grown on plastic surfaces have served the drug screening process well for decades, but it is now becoming apparent that 2D cell cultures are unable to accurately aid in the selection of clinically relevant drugs [20]. Three dimensional culture models bridge the gap between 2D cultures and animal models and are therefore ideally suited for phenotypic high-content screening (HCS). To investigate the differences between 2D and 3D cultures, we cultivated the same melanoma cell line (MCM DLN; 1000 cells/well) as 3D spheroids, with an initial diameter of approximately 200 µm. Over a period of 6 days, the morphology and cell growth of the melanoma spheroids seeded in non-adhesion microtiter plates were compared with cells seeded on a 2D plastic substrate. The cells in the spheroids exhibited an extraordinarily low proliferation rate, accompanied by a slower metabolic rate (Figure 1A–C). The MCM DLN spheroids reproduced the three-dimensional (3D) complexity and growth behavior of human melanoma much better than the 2D cultures (doubling time: >8 days for spheroids and 1–1.5 days for 2D cultures [21]). We next assessed the expression of the proliferation marker, Ki-67, using immunofluorescence staining. As shown in Figure 1D, small spheroids (1000 cells/spheroid) demonstrate a homogeneous Ki-67 positive cell distribution pattern, while in large spheroids (6000 cells/spheroid), Ki-67 positive cell staining is restricted to the outer cell layers, and the inner core regions remain dormant (Figure 1D; Appendix A). These data suggest that a proliferation gradient develops as multicellular tumor spheroids grow, which confirms the findings of Michel et al. [22].

### 2.2. Automation of 2D and 3D Cell-based Assays for Phenotypic Drug Screening

To maximize the amount of biologically relevant information that can be obtained from 2D cells cultured on plastic and 3D spheroid models, we harnessed a high-content analysis approach based on automated imaging (SpectraMax i3x Multi-Mode Microplate Reader, Molecular Device), using transmitted light and fluorescence microscopy combined with quantitative analysis to extract multiparametric data (Figure 1E and Appendix A). The measured parameters were the cell number in the 2D culture or the size of the 3D spheroids (area), viability (Presto blue), and proliferation (Ki-67).

### 2.3. Quercetin, Epigallocatechin Gallate (EGCG), and Resveratrol Display Different Effects on 2D and 3D Melanoma Cultures

In order to test quercetin (Figure 2A), 1000 MCM DLN, 1205Lu (metastatic), and MCM 1G (non-metastatic) cells were plated on 96-well plates either to attach and proliferate (2D) or to form spheroids (3D) for 4 days (Figure 2B). Then, the cells were treated with 50 to 0.4 µM of quercetin, with equivalent EtOH concentrations only, or left untreated for a further 3 days. The cell viability, cell number (2D) or spheroid area (3D), and cell proliferation (Ki-67) were detected (Figure 2, Figure 3 and Figure 4).

As depicted in Figure 2C–E, quercetin compromised the cell viability significantly in metastatic melanoma cells cultivated in 2D, compared to those cultivated in 3D cultures (treated cells were normalized to the equivalent EtOH concentrations and to the untreated control cells). On the contrary, in non-metastatic MCM 1G cells, there was no significant difference between the 2D and 3D models after quercetin treatment. Interestingly, low quercetin concentrations (0.4–12.5 µM) significantly increased the cell viability by up to 40% in MCM DLN and 25% in 1205Lu spheroids, compared to non-treated cells, while a relatively high concentration of quercetin (50 µM) decreased the cell viability by 20% to 60% (Figure 2C–E).

For further investigations, we performed a quick image analysis of the cell cultures (brightfield microscopy) to quantify the changes in cell number (2D) and the area of spheroids to observe their viability. By comparing the drug’s dose-response curves in the 2D cultures (Figure 2F–H) and 3D spheroids (Figure 2I–K), it was demonstrated that spheroids exhibit a lower sensitivity to quercetin. In the case of metastatic cells, a concentration range between 6.3 and 0.8 µM of quercetin boosted the growth of MCM DLN spheroids and 1205Lu 2D and 3D cultures over the 3-day treatment regime, whereas concentrations between 12.5 and 50 µM prevented cell growth (Figure 2F,G,I,J). In comparison, non-metastatic MCM 1G cultures (2D and spheroids) showed no enhanced cell growth, but they did show concentration-dependent cytotoxicity or growth-inhibiting activity, whereas the spheroids appeared to be less sensitive to the inhibiting effect of quercetin (Figure 2H,K).

On the contrary, epigallocatechin gallate (EGCG) and resveratrol caused only small viability differences between the 2D and 3D cultures, and in addition, the EGCG treatment showed no or just a very weak (>10 µM) effect on the cell cultures (Appendix A). MCM DLN cells cultured in 2D and treated with EGCG had a 35% reduction in cell survival, compared with untreated cells, while 3D spheroids treated with the same concentration of EGCG revealed no significant growth inhibition (Appendix A). In comparison to EGCG, resveratrol significantly reduced both the number of cells in 2D cultures and the area of 3D spheroids. Two dimensional cell culture treated with 50 µM of resveratrol showed a 50% decrease in the cell number, while 3D cells treated with the same concentration maintained a higher percentage of the covered well area (70%) (Appendix A). However, neither of the two substances boosted the growth of MCM DLN spheroids at low concentrations, as could be seen when they were treated with quercetin.

To verify and confirm the concentration-dependent pro- and anti-tumor activity of quercetin, MCM DLN spheroids (day 4) were treated with 0, 1, or 50 µM of quercetin, and the cell viability was determined by Presto blue for 24–120 h and compared to the control (0 h). Incubation with 1 µM of quercetin showed a time-dependent increase in the viability of the tumor spheroids (>30% after 5 days), whereas 50 µM showed the opposite effect (Figure 4A). Furthermore, significant activation of caspase 3 occurred when tumor spheroids were treated with more than 12.5 µM of quercetin, or 2D cultures were treated with more than 0.8 µM of quercetin for 48 h (Figure 3B).

To study the cell proliferation, the melanoma cells were fixed, permeabilized, and immunoassayed with Ki-67 antibody only or counter-stained with phalloidin (confocal microscopy) 3 days after treatment and subsequently probed using green and red fluorescence imaging and data analysis. As expected from the previously described experiments, a low concentration of quercetin (0.4 and 1 µM) slightly increased the number of proliferating cells in both MCM DLN and 1025Lu 3D cultures, but not in MCM 1G cells, compared to the control. At a high concentration (25 µM and 50 µM), proliferating cells were less abundant (Figure 4A–E).

These data clearly demonstrate the concentration-dependent pro- and anti-tumor activity of quercetin in human metastatic melanoma spheroids.

### 2.4. Quercetin Triggers Concentration-Dependent Decreased or Increased ROS Generation

Next, we assayed the reactive oxygen species (ROS) production to test whether different concentrations of quercetin influenced the generation of oxygen radicals in MCM DLN metastatic melanoma cells. At high concentrations (50 µM), quercetin enhanced the ROS generation in cells cultivated in 2D and 3D after 6 h of treatment. In comparison, 1 µM of quercetin had no effect in 2D cultures, but it significantly decreased the ROS generation in spheroids (Figure 5A,B). In both 2D and 3D cultures, the H_2_O_2_-activated intracellular ROS levels were significantly decreased or increased when the cells were co-cultivated with 1 µM or 50 µM of quercetin, respectively. Interestingly, the Nrf2 inhibitor N-acetyl cysteine (NAC) caused a significant decrease in the ROS level only in the 2D culture (Figure 5A), but not in the 3D spheroids (Figure 5B), after co-cultivation with 50 µM of quercetin. Together, these data suggest that quercetin shows both anti- and pro-oxidative effects.

### 2.5. Quercetin Demonstrates Concentration-Dependent Nrf2 Activation

Given that quercetin acts either as an antioxidant or pro-oxidant, with pro- and anti-tumor activity, in human metastatic melanoma spheroids, we surmised that quercetin might also show a concentration-dependent impact on oxidative stress-associated cell signaling molecules. To measure the downstream effects of quercetin-induced ROS in 2D and 3D melanoma cultures, we first established a MCM DLN cell line with a stable integrated Nrf2 reporter (Nrf2-Luc) using a lentiviral delivery system (see Materials and Methods, Appendix A). Next, the Nrf2-Luc MCM DLN cell line was cultured (2D and 3D) for 4 days and treated with 0–50 µM of quercetin for 24 h (Figure 6A). For spheroids treated with 0, 1, or 50 µM of quercetin, a time course was recorded over 5 days (Figure 6B). As depicted in Figure 6A, MCM DLN cells cultivated as 3D spheroids indicated a concentration-dependent increase of Nrf2 luciferase, whereas the 2D cultivation showed both a concentration-dependent increase and decrease when cells were treated with 0–3.2 µM of or 6.4–50 µM of quercetin, respectively. Interestingly, the Nrf2 signal in cells cultivated in 2D and treated with 0.8–3.2 µM was about twice as high as in spheroids when normalized to the non-treated control. Furthermore, treatment of spheroids with 1 µM of quercetin for 5 days demonstrated a time-dependent increase in the Nrf2 luciferase intensity (1.5 to 2.8-fold), compared to the control. However, spheroids treated with 50 µM peaked after 24 h and subsequently decreased to a 0.6-fold luciferase activity after 120 h (Figure 6B). Next, we evaluated the expression levels of the Nrf2 and Nrf2 target genes using RT-qPCR and western blotting after 24 h and 72 h of quercetin treatment (0, 1, and 50 µM). After 24 h, the expression of SOD, NQO1, GCLC, HmMOX, and Nrf2 gradually increased in quercetin-treated MCM DLN spheroids in a concentration-dependent manner (Figure 6C,D). After treatment with 1 µM of quercetin, the spheroids still showed either an enhanced mRNA level (SOD2, NQO1, and HmMOX) or the same level (GCLC and Nrf2) as the control (Figure 6C,D). In comparison, treatment with 50 µM of quercetin greatly reduced the gene and protein expression of the Nrf2 target genes (Figure 6C,D).

Because it has been shown that quercetin induces the upregulation of Nrf2, at least in part, via MEK/ERK1/2 signaling [23], we next investigated the ERK1/2 phosphorylation after quercetin treatment. As depicted in Figure 6E, the ERK1/2 phosphorylation is significantly increased when spheroids are treated with 1 or 50 µM of quercetin, compared to the control, after 24 h. Treatment for 72 h; however, strongly diminished the ERK1/2 phosphorylation when spheroids were treated with 50 µM of quercetin, whereas those stimulated with 1 µM were equal to the control (Figure 6E).

## 3. Discussion

In recent years, high-content screening (HCS) of anti-cancer drugs was mostly performed using conventional 2D cell cultures. However, drugs that showed high potencies against 2D cultures often failed in animal or cost-intensive clinical trials [20,24]. This can be, at least partly, attributed to the fact that 2D cultures do not accurately represent how cells grow or how they are affected by disease and injury in vivo, which often leads to misinterpretations in drug screening [24,25]. In vivo tumor proliferation occurs preferentially in well-nourished tissue regions close to the blood vessels, whereas nutrient-deprived and hypoxic tumor regions remain mostly dormant [26]. The cells in the hypoxic regions grow very slowly, are acidic, and have the tendency to be resistant to chemotherapy [27]. 3D models, like human tumor tissues, also contain heterogeneous zones of proliferating, quiescent, and dying cells [28,29] and show an expression pattern and spatial distribution of target molecules similar to that shown in vivo [25,30].

Thus, our study focuses on the comparison of 2D melanoma cultures with free-floating 3D spheroids, which, in the light of the abovementioned characteristics, appear to have higher predictivity regarding the antiproliferative effect of anti-cancer drugs than 2D cultures. Moreover, they are also suitable for high-content screening (HCS). By comparing 2D melanoma cultures with 3D spheroids, we found that cell growth and metabolism is much slower when cells are cultivated as 3D spheroids, with a cell doubling time that is five to eight times slower than that in 2D cultures. The spheroids generated here, with a diameter of 200–350 µm, can be considered as simplified models of tiny microtumors, in which cancer structures can organize into a hierarchical arrangement, with properties and characteristics found in human melanoma tumors in vivo. Hence, it is an anticipated consequence that 3D spheroids behave differently from 2D cell cultures upon treatment with anti-cancer substances.

In the present study, we focused on quercetin, as it has been described as a substance with many anti-carcinogenic properties, including the downregulation of cell proliferation and anti-apoptotic proteins, the inhibition of mutant p53 or tyrosine kinase, or free radical scavenging activity [31,32,33,34]. Epidemiological studies have also shown that quercetin reduces the risk of colon and gastric cancer [35,36].

Among other differences (Figure 2), a remarkable difference could be detected between 2D and 3D when cell cultures were treated with different concentrations of quercetin. We could determine that higher concentrations (>12.5 µM) of quercetin reduce melanoma growth in all three cell lines, whereas low concentrations (6.3 µM–0.4 µM) indicate the opposite effect in MCM DLN and 1205Lu 3D spheroids (Figure 2, Figure 3 and Figure 4) but not in MCM 1G (non-metastatic melanoma) cells. These effects could be seen in MCM DLN only when cells were cultivated in 3D, whereas in 1205Lu cells, they were observed in both 2D and 3D cultures. High (>12.5 µM) or low concentrations decreased or increased cell viability, spheroid size, and cell proliferation (Ki-67), respectively. Furthermore, MCM DLN cells cultivated in 2D already showed significant caspase 3 activity and a strong toxic effect at very low quercetin concentrations (>0.4 µM), whereas in 3D spheroids, apoptotic cells could only be detected in quercetin concentrations ≥12.5 µM (Figure 4). Consistent with our findings, Robaszkiewicz et al. demonstrated that low concentrations of quercetin promote cell proliferation in A549 (human lung carcinoma) cultures and increase the total antioxidant capacity of cells, and higher concentrations had the opposite effect, which is most likely due to the reduced intracellular ROS level and increased expression of endogenous antioxidant enzymes [13]. Lee et al. also showed that a quercetin dose >60 µM leads to cytotoxicity, but a low dose may provide an advantage for mesothelioma cell line survival through quercetin-induced Nrf2 activation and, subsequently, the ARE-mediated expression of a battery of cytoprotective genes. Furthermore, there is abundant evidence that the activation of Nrf2 can either suppress carcinogenesis, especially in its earliest stages, or promote tumorigenesis by protecting cancer cells. Besides the genetic mutations in Keap1 and/or Nrf2, which are rather rare, it was shown that common oncogenes, such as MYC, KRAS, and BRAF, enhance Nrf2 expression [37]. Thus, oncogenes may promote tumorigenesis at least partially through the Nrf2-dependent cytoprotective activity, including the reduction in ROS levels and the establishment of a more beneficial intracellular environment for the survival of tumor cells. These findings suggest that the tumor promoting effect in our melanoma models can also be attributed to a quercetin-induced upregulation of Nrf2/ARE-mediated cytoprotective genes in a concentration- and duration of exposure-dependent manner, depending on the tumor stage and cultivation conditions.

Therefore, next, we investigated the pro- and/or anti-oxidative properties of quercetin and the underlying signaling pathways, Nrf2, MAPK, and NF-κB, in a dose- and time-dependent manner in melanoma spheroids. By measuring the reactive oxygen species (ROS) activity in MCM DLN tumor cultures, we demonstrated a concentration-dependent decrease or increase of ROS production in 3D spheroids, but not in 2D cultures (Figure 5). A similar effect could also be observed when melanoma cultures were co-treated with H_2_O_2_. Again, a high concentration further increased the ROS production, whereas low concentrations (1 µM) diminished this effect, suggesting that the anti- and pro-oxidative activity of quercetin is concentration-dependent. Recently, it was demonstrated that 3D-grown cancer cells showed increased levels of ROS production, compared to 2D cultures, and that moderate ROS levels promote cell survival and proliferation [38,39]. Nrf-2 analysis revealed a concentration-dependent increase of Nrf2 reporter activity and corresponding transcriptional targets, SOD2, NQO1, GCLC, and HmMOX1, in MCM DLN spheroids when treated with quercetin for 24 h, compared to the control (Figure 6). Interestingly, after 24 h, the Nrf2 activity continuously decreased, reaching a lower concentration than the control after 5 days. In contrast, a low dose of quercetin (1 µM) showed just a time-dependent increase in the Nrf2 activity and Nrf2 target expression. These data indicate that a low concentration of quercetin exerts a protective effect through radical-scavenging mechanisms, together with the concomitant activation of Nrf2 signaling. On the contrary, high concentrations of quercetin might potentiate oxidative stress, which allows the threshold for cell death to be reached. In addition to the role of ROS in Nrf2, several protein kinases, such as ERK and NF-κB, can be activated by quercetin and have been recognized as the key regulators of the upstream signaling involved in Nrf2 activity [23]. Furthermore, Lee et al. found that quercetin can bind directly to RAF and MEK protein kinases and inhibit their phosphorylating activities [40]. We found that quercetin treatment significantly increased ERK and NF-κB activation after 24 h, but this effect was reversed after 3 days when spheroids were treated with high, but not low, concentrations of quercetin. Because a sustained ERK phosphorylation is necessary for cell proliferation and survival, high concentrations (50 µM) of quercetin contribute to an increased occurrence of cell death, whereas low concentrations promote cell proliferation and survival in metastatic melanoma spheroids.

Quercetin has been suggested as a complementary approach to treating melanoma. The biphasic nature of quercetin has been described by others as having the potential to be used as a dietary component for both the prevention of melanoma at low doses and as adjuvant therapy for metastatic melanoma at high doses [18,19,41]. Here, we clearly demonstrate that quercetin shows both pro- and anti-carcinogenic activity in melanoma cells, depending on the quercetin concentration, the way the cells are cultivated, and the stage of the melanoma cells. Therefore, more research using physiologically relevant test models is needed, especially to better understand the relationship between the tumor stage and the different modes of action of quercetin in melanoma to maximize tumor toxicity and avoid any tumor-promoting activity or side effects from the antioxidant properties of quercetin.

## 4. Materials and Methods

### 4.1. Cell Culture

1205Lu cells (ATCC^®^ CRL-2806), MCM 1G, and MCM DLN (kindly provided by P. Petzelbauer, Department of Dermatology, Medical University of Vienna, Austria) were cultured in a melanoma isolation media (MIM) at 37 °C in a humidified atmosphere containing 5% CO_2_. The MIM medium was prepared by mixing 14.08 g of an MCDB 153 medium (M7403, Sigma-Aldrich) in 800 mL of ddH2O, followed by the addition of 12.6 mL of sodium bicarbonate (7.5%) (S8761; Sigma-Aldrich), with the adjustment of the pH to 7.4 and sterile filtration. Furthermore, 200 mL of an L-15 (Leibovitz) medium (L1518, Sigma-Aldrich), 20 mL of FCS (10270-106, ThermoFisher Scientific, Waltham, MA, USA), 10 mL of Penicillin/Streptomycin (15140-122, ThermoFisher Scientific), 0.5 mL of Insulin (I9278, Sigma-Aldrich), 1680 µL of calcium chloride solution (21115, Sigma-Aldrich), 2µg/mL of Ciprofloxacin (17850-5G-F, Sigma-Aldrich), and 50 ng/mL of EGF (53003-018, ThermoFisher Scientific) were added. Cells were passaged every 3–5 days before reaching 80% confluency. For the 3D cell culture, PrimeSurface 96U plates (MS-9096UZ, S-bio) and an InSphero GravityPLUS System (InSphero, Schlieren, Switzerland; PerkinElmer, Waltham, MA, USA) were used, unless otherwise indicated. The spheroids were generally grown 4 days before being used for the experiments.

### 4.2. Immunofluorescence Staining

The spheroids were chemically fixed using 4% paraformaldehyde (15670799, ThermoFisher Scientific) for 30 min. Subsequently, the cells were permeabilized using 0.2% Triton X-100 (11332481001, Sigma-Aldrich) for 15 min. After washing, the cells were stained with phalloidin (A12379, ThermoFisher Scientific) and Ki-67 FITC (33-4711, Invitrogen, Carlsbad, CA, USA). Antibodies were used at 1:800 and 1:500 dilutions, respectively, and allowed to incubate at 4 °C overnight. A Hoechst33342 solution (PA-3014, Cambrex Bioscience, East Rutherford, NJ, USA) was applied at a 1:10.000 dilution for 30 min, before the spheroids were finally washed and mounted on a microscopy slide in collagen I (A10483-01, Gibco, Gaithersburg, MD, USA). A frame (Ab0576, ThermoFisher Scientific) was applied to the slide before mounting in order to prevent the spheroid from being compressed. Subsequently, the slides were left to dry before being subjected to confocal imaging using a Leica TCS SP8 laser scanning microscope.

For quantitative analysis using the SpectraMax i3x Multiplate Reader, the cells were fixed and permeabilized, as described above. For background measurement, each well was analyzed before the staining. Then, cell tracker deep red dye (C34565, Invitrogen) was added at a 1:5000 dilution for 20 min. After washing, the cells were stained with Ki-67 FITC at a 1:500 dilution and allowed to incubate at 4 °C overnight. On the next day, microscopic images were acquired at wavelengths of 456/541 nm Excitation/emission (Ex/Em) for Ki-67 FITC and 627/713 nm (Ex/Em) for the cell tracker, and the fluorescence intensity was quantitatively assessed at an Ex/Em of 490/525 for Ki-67 and 630/660 nm for the cell tracker using the SpectraMax i3x Multiplate Reader.

### 4.3. Viability

Viability was determined with a presto blue assay (A13262, Invitrogen) using the Spectra Max i3x Multiplate Reader and Transmitted Light (TL) detection cartridge (5022671, Molecular Devices, San Jose, CA, USA). Presto blue was added directly to the wells to yield a 1:10 dilution and allowed to incubate at 37 °C for 60 min. Measurements were conducted at an Ex/Em of 555/585 using the SpectraMax i3x Multiplate Reader. For samples with substance treatment, an empty medium control and treatment was used as a background measurement for each individual well to compensate for potential autofluorescence of the substance.

### 4.4. D Culture Cell Count and 3D Area Measurement

The number of cells in 2D cultures was determined using the object count feature of the SpectraMax i3x MiniMaxTM 300 Imaging Cytometer (5022671, Molecular Devices). Images for the object count were acquired using transmitted light and optimized settings for each cell line.

Area measurements were conducted using the MiniMaxTM 300 Imaging Cytometer (5022671, Molecular Devices), which is part of the SpectraMax i3x. Images of the spheroids were acquired before and after treatment in order to determine the change in size. FIJI [42] and the FIJI macro INSIDIA [43] were used to achieve spheroid segmentation. The “Analyze Spheroid Cell Invasion In 3D Matrix” macro in FIJI was subsequently used to measure the area of the segmented spheroid images. The unit of output received was pixels, with a dimension of 1.9 × 1.9 µm^2^. The change of each individual spheroid, relative to the same spheroid before treatment, was calculated. Furthermore, the change in the area, relative to the geometric mean of the controls without treatment within the same experiment, was calculated.

### 4.5. Substance Treatment

The cells were treated with different concentrations of resveratrol (R5010, Sigma-Aldrich), Epigallocatechin gallate (EGCG, E4143, Sigma-Aldrich), and quercetin (Q4951, Sigma-Aldrich) dissolved in ethanol (EtOH). The highest concentration of EtOH (0.18%, 0.07%, and 0.05% for quercetin, EGCG, and resveratrol, respectively) applied to the cells during treatment was used as a control in order to eliminate EtOH effects from our data. Furthermore, the data for quercetin were normalized to an EtOH curve, with the same concentrations as those used during the substance treatment, to further reduce the effect of the solvent.

### 4.6. ROS Assay

The cell monolayers and spheroids were washed with 1xPBS (10010-023, ThermoFisher Scientific) and permeabilized with 20 µM of 2′,7′-Dichlorofluorescin diacetate (DCFDA) (D6883-50MG, Sigma-Aldrich) for 30 min at 37 °C and 5% CO_2_. After washing the cells with 1 × PBS twice, they were treated with 1 µM of quercetin, 50 µM of quercetin, 1 mM of hydrogen peroxide (BP2633500, ThermoFisher Scientific), 1 mM of hydrogen peroxide; 1 µM of quercetin, 1 mM of hydrogen peroxide, and 50 µM of quercetin; 5 mM of N-Acetyl-L-cysteine (A9165-5G, Sigma-Aldrich), 5 mM of N-Acetyl-L-cysteine, and 1 µM of quercetin; 5 mM of N-Acetyl-L-cysteine and 50 µM of quercetin; or they were left untreated for 60 min at 37 °C and 5% CO_2_. The oxygen-reactive species were quantified by measuring the fluorescence at 485 nm (excitation) and 535 nm (emission) using the transmitted light detection cartridge (5022671, Molecular Devices) in the SpectraMax i3x platform and normalized to the cell count or area, which were determined using the Mini MaxTM 300 Imaging Cytometer (5024062, Molecular Devices). The treatments were performed in sextuplicate, and the fold changes of the different treatments, relative to the untreated control sample, were calculated. The evaluated data were graphically processed in Excel.

### 4.7. Luciferase Reporter Assay

The firefly luciferase reporter was detected by applying the ONE-GLOTM Luciferase Assay System (E6120, Promega, Madison, WI, USA). The Relative Luminescence Units (RLU) were measured using the SpectraMax i3x Luminescence Glow cartridge (Lum 384) (0200-7015POS, Molecular Devices) and normalized to the cell count or area, which were determined using the Mini MaxTM 300 Imaging Cytometer (5024062, Molecular Devices). The treatments were performed in sextuplicate, and the fold changes of the different treatments, relative to the untreated control sample, were calculated.

### 4.8. Protein Lysates and Immunoblots

The spheroids were lysed in situ in a hot Laemmli sample buffer (1610737, Bio-Rad, Hercules, CA, USA) containing 5% ß-mercaptoethanol (M7522-100ML, Sigma-Aldrich) and further incubated for five heating-freezing cycles at 95 °C for 5 min and in liquid nitrogen for 1 min. The total protein extracts were separated using 7.5% Mini-PROTEAN^®^ TGXTM Precast Protein Gels (456-1023, Bio-Rad) and transferred onto a nitrocellulose membrane using the Trans-Blot^®^ TurboTM Blotting System (1704155, Bio-Rad). The proteins were blotted using the preset transfer protocol, “Mixed WM” (25 V, 2.5 A, 7 min). The membranes were blocked with 5% non-fat dry milk (9999S, New England Biolabs) in 0.1% (*v*/*v*) Tween-20 (P7949-100ML, Sigma-Aldrich) diluted in 1 × PBS (10010-023, ThermoFisher Scientific). The following primary antibodies were used: recombinant anti-Nrf2 antibody (EP1808Y) (ab62352, Abcam, Cambridge, UK), SOD-2 (E-10) (sc-137254 HRP, Santa Cruz, CA, USA), NQO1 (H-9) (sc-376023 HRP, Santa Cruz), recombinant anti-ERK1 (phospho T202) + ERK2 (phospho T185) antibody (EPR19401) (ab201015, Abcam), anti-ERK1 + ERK2 antibody (ERK-7D8) (ab54230, Abcam), ß-actin antibody (C4) (sc-47778, Santa Cruz), and vinculin antibody (7F9) (sc-73614 HRP, Santa Cruz). The secondary antibodies were purchased from Cell Signaling: anti-rabbit IgG, HRP-linked antibody (7074S) or anti-mouse IgG, and HRP-linked Antibody (7076S). The immunoblot was developed by applying the Clarity Western ECL Substrate (1705060, Bio-Rad), according to the manufacturer’s instructions. The proteins were quantified via chemiluminescence imaging using the ChemiDoc MP platform (17001402, Bio-Rad) and in silico analyzed via the Image Lab 6.0.1 Software. The evaluated data were graphically processed using Excel.

### 4.9. RNA Isolation, cDNA Synthesis, and Real-time PCR (RT-qPCR)

The RNA of the spheroids (6000 cells/well in 96 U plate) treated with quercetin (0 µM, 1 µM, and 50 µM) for 24 h or 72 h was isolated by applying the RNeasy^®^ Mini Kit (74104, Qiagen, Hilden, Germany), according to the manufacturer’s instructions. The cDNA was synthesized using the qScriptTm cDNA Super Mix (84034, Quantabio, Beverly, MA, USA), according to the manufacturer’s instructions.

The Nrf2 gene expression and various Nrf2 target gene expressions (SOD2, NQO1, GCLC, and HmMOX1) were assessed using RT-qPCR with pre-designed TaqMan^®^ Gene Expression Assays (Hs00975961_g1 NFE2L2, Hs00167309_m1 SOD2, Hs01045993_g1 NQO1, Hs00892604_m1 GCLC, Hs01110250_m1 HMOX1, and Hs03929097_g1 GAPDH), consisting of a pair of unlabeled PCR primers and a TaqMan^®^ probe with a FAM dye label on the 5′ end and minor groove binder and nonfluorescent quencher on the 3′-end. The RT-qPCR reaction mix contained a final volume of 10 µL of TaqMan^®^ Gene Expression Master Mix (4369514, ThermoFisher Scientific), 1 µL of TaqMan^®^ Gene Expression assay, 5 µL of nuclease free water (AM9937, Ambion, Austin, TX, USA), and 4 µL of cDNA template diluted 1:10. The RT-qPCR was performed in triplicates using the QuantStudio 7 Flex (278671303, Applied Biosystems, Foster City, CA, USA), applying the following cycling conditions: 2 min at 50 °C for UDG incubation and 10 min at 95 °C for enzyme activation, followed by 45 cycles of denaturation for 15 s at 90 °C and annealing/extension for 60 s at 60 °C. The data were analyzed using QuantStudioTM Real-Time PCR Software v.1.3. The expression levels of the Nrf2 and Nrf2 target genes were calculated according to the comparative Cq method (2-ΔΔCT), with GAPDH as the reference gene.

### 4.10. Statistics

The data analysis was performed using the Real Statistics Resource Pack software (Release 5.4), copyright of Charles Zaiontz (2013–2018). The boxplots show the maximum and minimum values, as well as the median and second and third quartile. The column charts are always shown with the ± standard deviations (S.D.).

## Figures and Tables

**Figure 1 molecules-26-00717-f001:**
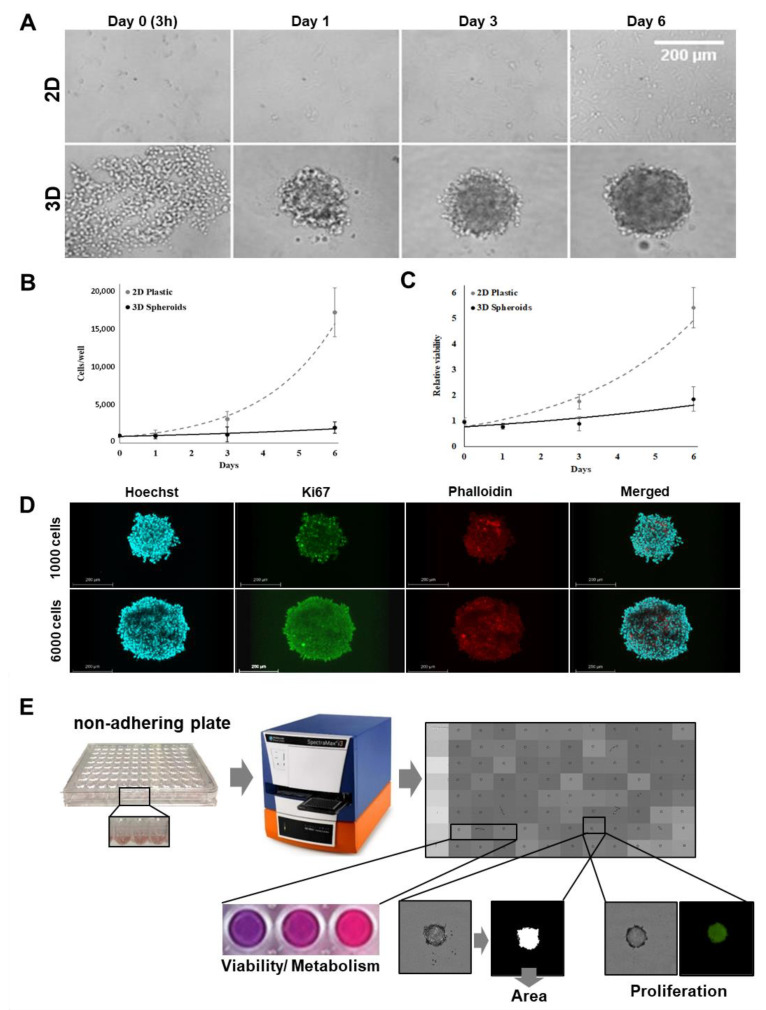
Growth kinetics of MCM DLN human melanoma cells in 2D and 3D cultures. (**A**) One thousand cells were seeded to generate 2D cultures (adherent plastic plates) and 3D spheroids (non-adherent plates), and cell growth was detected by phase-contrast microscopy on day 0 (3 h after seeding), 1, 3, and 6 using a Leica DMI6000B inverted microscope. (**B**) On days 1, 3, and 6, cells were harvested by trypsinization to determine the cell number. (**C**) Cell viability measurements were conducted using Presto blue, normalized to the viability of day 0 (3 h after seeding). Each data point represents the mean ± standard deviation (S.D.) of six independent experiments. (**D**) One thousand or 6000 MCM DLN cells were cultured as three-dimensional (3D) cell aggregates using non-adherent plates. After 4 days, the spheroids were fixed, permeabilized, and stained with antibodies for the proliferation marker, Ki-67, Hoechst, and phalloidin. Images were acquired using a Leica TCS SP8 confocal laser scanning microscope. It should be noted that Ki-67 positive cells in the spheroids seeded with 1000 cells are found to be evenly distributed and, in the larger spheroids (6000 cells), only in the outer rim. Scale bars: 200 μm; Z-stack: Appendix A. (**E**) Schematic presentation of the high content phenotypic drug screening assay design.

**Figure 2 molecules-26-00717-f002:**
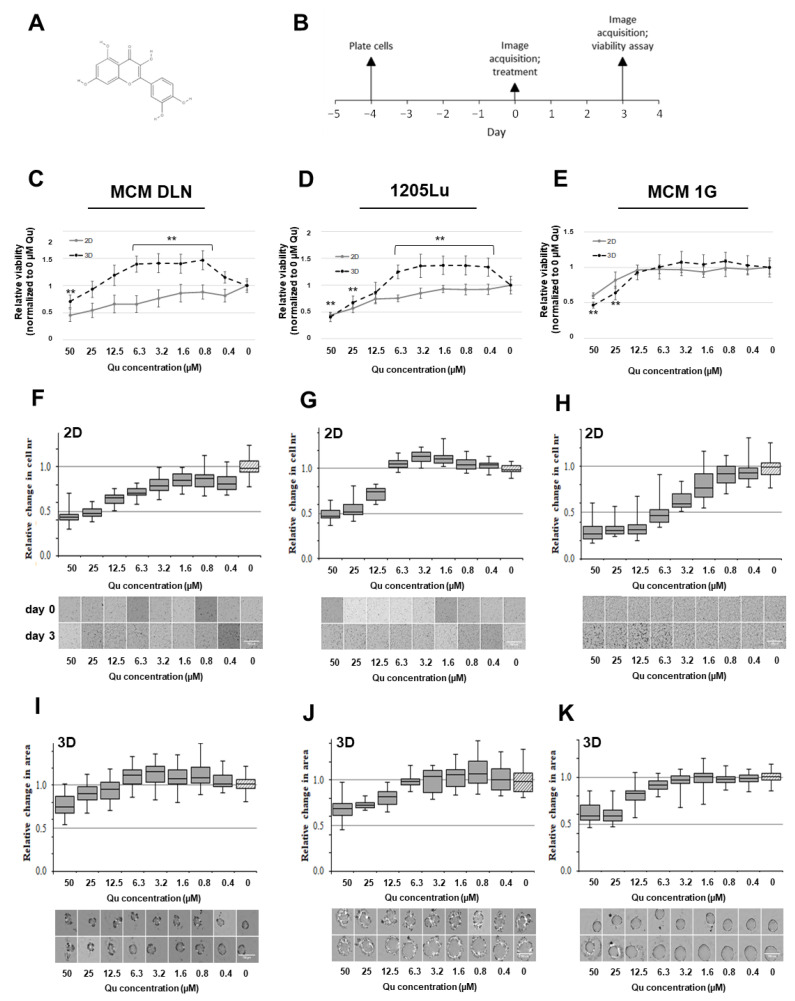
Effect of quercetin on cell viability and cell growth in 2D and 3D human melanoma cell lines. The 2D cell culture on plastic and melanoma spheroids on day 4 were treated with 50–0.4 µM of quercetin, with an equivalent EtOH concentration only, or left untreated. The metabolic activity (viability), cell number (2D), and spheroid area were evaluated 3 days later. The data are normalized to the equivalent solute concentration and to the non-treated cells. (**A**) Chemical structure of quercetin. (**B**) Timeline of the 2D and 3D spheroid preparation, treatment, and imaging. (**C**–**E**) Presto blue (cell viability) measurement in 2D and 3D human melanoma cells after quercetin treatment. (**C**) MCM DLN, (**D**) 1205Lu, and (**E**) MCM 1G melanoma cells. The data are expressed as the mean ± S.D. performed at least 4 times in octuplicates. The *p*-values are expressed as ** *p* < 0.01, compared to the control (0 µM). Evaluation of the cell number (**F**–**H**) and the spheroid area (**I**–**K**) of MCM DLN (F, I), 1205Lu (**G**,**J**), and MCM 1G (**H**,**K**) melanoma cells. The data are presented as a box-whisker-plot (graphics) performed at least 4-times in octuplicates and in corresponding images (below).

**Figure 3 molecules-26-00717-f003:**
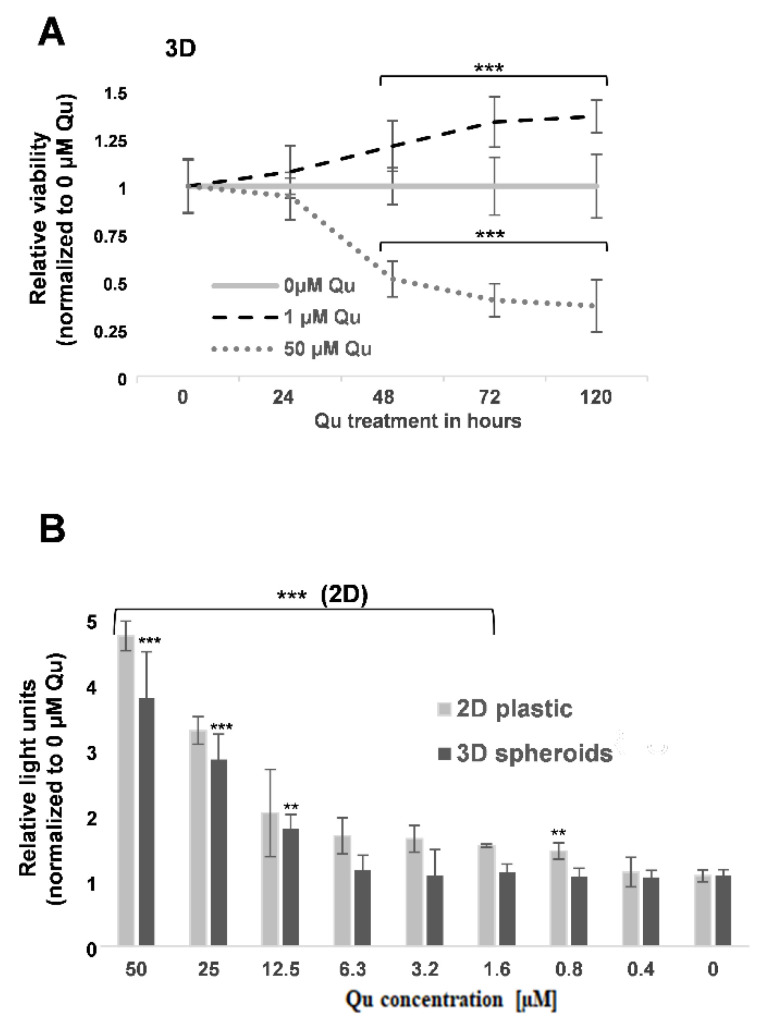
Effect of quercetin on cell viability in 3D human melanoma cell lines. (**A**) MCM DLN spheroids were treated on day 4 with 0, 1, or 50 µM of quercetin, and the cell viability was assessed every day for five days using Presto blue. (**B**) Two dimensional cell culture on plastic and melanoma spheroids on were treated day 4 with 50–0.4 µM of quercetin or left untreated, and the Caspase 3 activity (luminescence) was measured 24 h later. The data are normalized to the cell number (2D) or spheroid area (3D), the equivalent solute concentration, and to the non-treated cells. The experiments were performed at least four times in sextuplicates and are expressed as the mean ± S.D. The *p-*values are expressed as ** *p* < 0.01, and *** *p* < 0.001, compared to the control (0 µM).

**Figure 4 molecules-26-00717-f004:**
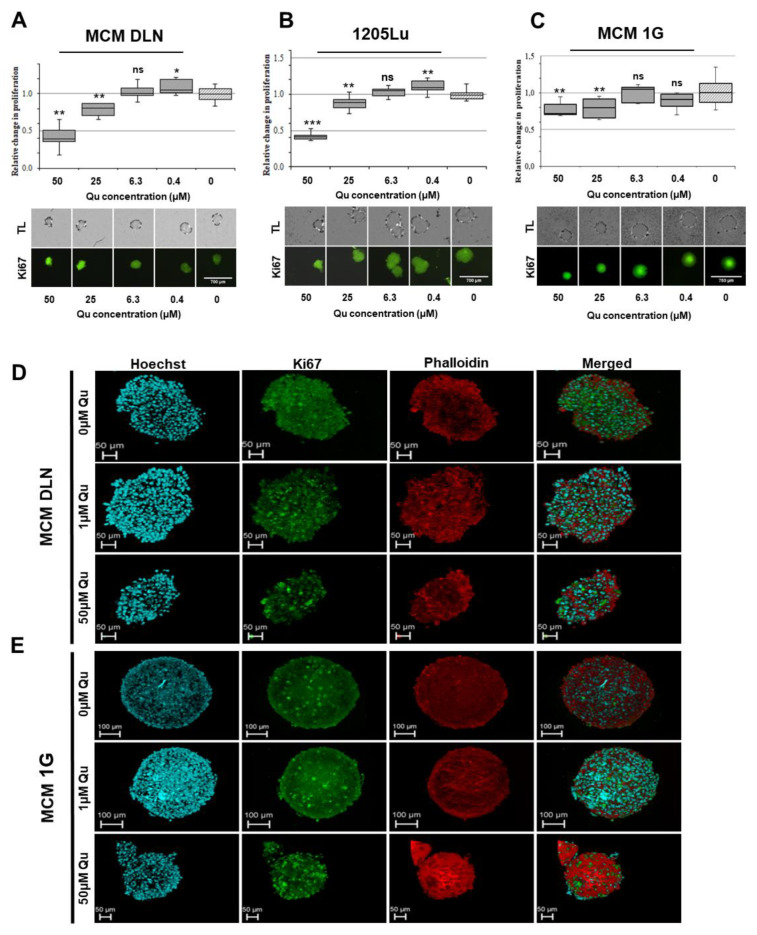
Effect of quercetin on cell proliferation in 3D human melanoma cell lines. Spheroids were treated on day 4 with 50–0 µM of quercetin, and the cell proliferation was assessed three days later using Ki-67 antibody (green). The experiments were performed at least three times in octuplicates. The data are normalized to the equivalent solute concentration and to the non-treated cells. (**A**) MCM DLN, (**B**) 1205Lu, and (**C**) MCM 1G spheroids. The *p-*values are expressed as * *p* < 0.05, ** *p* < 0.01, and *** *p* < 0.001, compared to the control (0 µM). Melanoma spheroids (seeded with 1000 cells) were treated for 3 days with 1 µM or 50 µM of quercetin or left untreated and stained with Ki-67, Hoechst, and phalloidin. Confocal microscopic images from (**D**) MCM DLN and (**E**) MCM 1G melanoma were acquired using a Leica TCS SP8 confocal laser scanning microscope. Scale bars: 50 or 100 μm; Z-stack: Appendix A.

**Figure 5 molecules-26-00717-f005:**
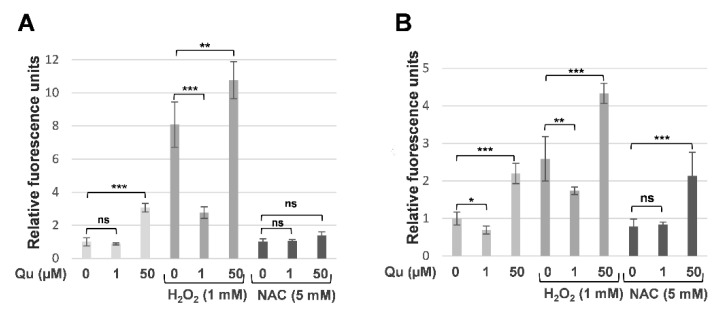
Intracellular ROS activation in 2D and 3D human melanoma cells, after quercetin treatment. 2D cell culture (**A**) and melanoma spheroids (**B**) were treated on day 4 with 0, 1, or 50 µM of quercetin alone or together with 1 mM of H_2_O_2_ or 5 mM of NAC for 6 h, with the pro- and/or anti-oxidative activity measured using 2′,7′-Dichlorofluorescin diacetate (DCFDA) (D6883-50MG, Sigma-Aldrich, St. Louis, MO, USA). The experiments were performed three times in sextuplicate. The data are normalized to the equivalent solute concentration and to the non-treated cells and are expressed as the mean ± S.D. The *p-*values are expressed as * *p* < 0.05, ** *p* < 0.01, and *** *p* < 0.001, compared to the control (0 µM).

**Figure 6 molecules-26-00717-f006:**
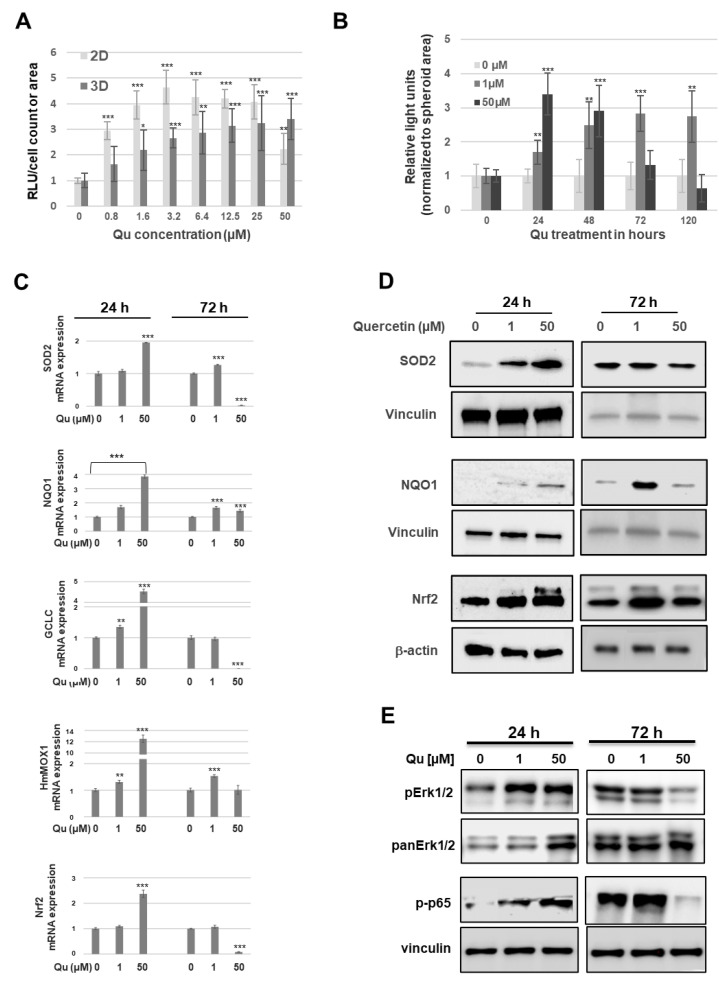
Nrf2 activation in human melanoma cell after quercetin treatment. (**A**) One thousand MCM DLN cells cultured in 2D or 3D spheroids were treated with quercetin (0–50 µM) and Nrf2 reporter (luciferase) and measured 48 h later. (**B**) Spheroids were treated with 0, 1, or 50 µM of quercetin, and the Nrf2 luciferase activity was measured after 0, 24, 48, 72, and 120 h. (**C**) The expression level of the Nrf2 (NFE2L2) and Nrf2 target genes (SOD2, NQO1, GCLC, and HMOX) in melanoma spheroids were detected by RT-qPCR 24 h and 72 h after quercetin treatment (0, 1, or 50 µM). The *p-*values are expressed as * *p* < 0.05, ** *p* < 0.01, and *** *p* < 0.001, compared to the control (0 µM). (**D**) Western blotting using antibodies against SOD2, NQO1, and Nrf2 or (**E**) pERK1/2 and panERK1/2.

## Data Availability

Data presented in this study are available on request from the corresponding authors.

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
