# Peer review of "Concentration-Dependent Pro- and Antitumor Activities of Quercetin in Human Melanoma Spheroids: Comparative Analysis of 2D and 3D Cell Culture Models"

_molecules, 2021, doi:10.3390/molecules26030717_

Round 1

Reviewer 1 Report

The manucript established stable and reproducible method for the 3D spheroid , and different sensitivities in 2D and 3D melanoma models were observed. The experiments are well performed and the findings provide useful information. I recommend for publication after clarifying several concerns .

 (1) Your results show well that quercetin has concentration-dependent pro- and antitumor activities, and 3D model is less sensitive. What is the significance of these two findings? Does it have inspiration for the application of quercetin in melanoma? It’s suggested to add your opinion in the DISCUSSION.

(2) If I understand correctly, you concluded that low concentrations do not induce the pro-tumor effect in metastatic melanoma spheroids in the non-metastatic cell line. However, Figure 2E cannot support the conclusion well. For the 3D spheroid of non-migrating MCM G1 (Figure 2E), the viability of spheroid s treated with 0.4~3.2 μM quercetin is also greater than spheroid s treated with 0 μM, although with no statistical significance. Assuming that the conclusion is true, what is the potential inspiration for us?

(3) Figure 6 shows elegant data. When treated with low concentration (1 μM) of quercetin, Nrf2 and target genes are continuously activated (24h~72h); when treated with high concentration (50 μM) of quercetin, Nrf2 and target genes are only activated for a short time (peak at 24h). However, it is a little arbitrary to attribute the tumor-promoting and tumor-suppressing effects of quercetin directly to Nrf2. The effect is possibly related to Nrf2. The expression words need to be changed.

  1. Other concerns

 (1) The immunofluorescence staining section in Methods does not mention optical clearing. The penetration depth of the laser scanning confocal microscope is limited to about 100 μm. Larger samples need to be optically cleared to make the sample transparent, otherwise, the cross section will be brighter at edge and darker inside. If you have not operate optical clearing, “the Ki-67 positive cells staining is restricted to the outer cell layers (Line 122)” that you observed may not be consistent with the actual situation. I hope you perform 3D reconstruction of z-stack images or inspect x-z orthogonal planes to ensure that cells at depths >100um can also be imaged well. Although your Video_S1 provides a 3D reconstruction video, I look forward to a 360° drag-and-drop version.

(2) In Figure 2F, there is a gap between 0.4 μM and 0 μM. The data cannot connect smoothly. May you explain why?

(3) After resveratrol treatment, the number of 2D cells decreases to 50%, and the area of ​​3D spheroid decreases to 70% (Line 256). However, it does not seem to be correct to conclude that 2D cultures is more sensitive. For 2D cells, the area is proportional to the number of cells, area=k1*n. For 3D spheroid, the volume is proportional to the number of cells, volume=4/3*π*r^3=k2*n. If we calculate the number of cells for 2D, we should better calculates r^3 for 3D spheroid, which will make it more fair to compare the 2D and the 3D.

(4) “MCM DLN cells cultured in 2D and treated with EGCG had less than 35% survival compared with untreated cells (Line 253)”. Does it mean survival <35%? From the figure, it seems that the survival reduced by 35%.

(5) Figure 3 and Figure 4 are quoted incorrectly in the text.

(6) “1 µM quercetin had no effect in 2D cultures but significantly decreased ROS generation in spheroids (Line 339).” But in fact, both 2D cultures and 3D cell spheres have similar trends. In 2D cultures, 1 μM quercetin treatment also reduced ROS, even if it was not statistically significant, which may be due to the within-group deviation.

(7) In Figure 5B, the ROS of the 50 μM Qu + NAC group is higher than that of the 0 μM Qu + NAC group. This difference is not observed in Figure 5A. Does the pro-oxidation effect of Qu become more obvious in 3D? Would you explain the reason for the difference?

(8) Axis of some figures, such as Figure 2F, is showed incompletely.

Reviewer 2 Report

Hundsberger et al. report about „Concentration dependent pro- and antitumour activities of quercetin in human melanoma spheroids: Comparative analysis on 2D and 3 cell culture models”

General comments:

The study is well designed and the reader can easily follow the paper and the argumentation of the authors. There are only a few formatting errors and typos that have to be corrected:

  1. Line 15: indentation of affiliation 3
  2. Formulation of lines 30 to 32: it is an unfortunate wording to state “…the results clearly confirm that the role …is still controversial”. It is highly recommended to rephrase this sentence because the data provided clearly show a concentration depency of the 2D MCM DLN and 1205Lu system as well as in the MCM DLN spheroid system.
  3. Lines 44-49: delete
  4. Figure 2: In the Materials and Methods section the cell lines are introduced as 1205Lu and MCM1G, please correct in the figure.
  5. Either the explanation of Figure 1E is missing in the figure caption or just the “(E)” is missing.
  6. Line 313: increase
  7. Line 324: three
  8. Line 325: 3 times
  9. Line 359: cell
  10. Line 363: was cultured
  11. Line 421: Replace “fabricated” (generated?)
  12. Line 431: reduces

Specific comments:

  1. Presto Blue is a Resazurin derivate that can be used to judge viability but not “metabolism” in its strict meaning. Therefore, it is recommended to remove “metabolism” in all those descriptions where just viability measurements were made. For the assessment of the overall metabolism of a cell more parameters need to be detected.
  2. Figure 6A is missing statistical analysis and/or the labels of the analysis results
  3. Figure 6C and D unfortunately display only the PCR and Western blot results of the 3D culture. It would be highly desirable to see the 2D results of this part for better comparison.
  4. Figure 6C: the y-axis are not legible, please enlarge the numbers.
  5. Figure S1: Please include an example of a spheroid segmentation result.